# CATEGORICAL TRACE LOOP NETWORKS FOR GAUGE-RANDOMIZED HOLONOMY REGRESSION

**Yoshihiro Maruyama**
Department of Mathematical and Information Sciences
Kyoto University
Kyoto, Japan
`maruyama@i.h.kyoto-u.ac.jp`

## ABSTRACT

Gauge ambiguity is a pervasive obstacle when learning from group-valued transport data: edge measurements depend on arbitrary local coordinate choices, while the quantities we care about are gauge-invariant. Inspired by lattice gauge theory, where meaningful observables are built from loop holonomies rather than individual edge variables, we study a $SO(3)$ learning problem on a discrete torus with random vertex-wise gauges. We adopt a categorical viewpoint in which an edge connection defines holonomy functorially on edge-paths of the 1-skeleton; in the flat/noiseless regime, this holonomy descends to the fundamental groupoid of the torus cell complex. Gauge transformations act as natural isomorphisms. This viewpoint leads us to Categorical Trace Loop Networks (CTLN): a novel architecture that learns from loop- and face-based gauge invariants obtained by functorial holonomy composition and trace/angle scalarization. On gauge-randomized torus holonomy regression, CTLN achieves a test MAE of $0.1747$, while a standard message passing network and a spectral connection-Laplacian baseline both remain near $0.81$ MAE. These results show that in gauge-dominated regimes, learning on categorical invariants capturing global topology and higher-order consistency provides a highly effective method.

## 1    INTRODUCTION

Many modern learning problems involve *group-valued transport* data: measurements on edges encode how local coordinate frames, poses, or latent states transform across a discretized domain. A difficulty here is that the raw representation is typically *gauge-dependent*: a vertex-wise change of coordinates conjugates each edge transformation while leaving the underlying physical or geometric content unchanged. This phenomenon is central in lattice gauge theory, where gauge-invariant observables are built from loop holonomies (Wilson loops) rather than individual edge variables (Wilson, 1974; Kogut, 1979).

In machine learning, analogous gauge effects arise whenever the choice of local frames is arbitrary or latent, and successful prediction requires respecting the induced invariances. A standard response is to enforce symmetry through equivariant architectures. Group-equivariant convolutional networks (Cohen & Welling, 2016) and their gauge-equivariant extensions (Cohen et al., 2019) exemplify how encoding symmetry in the model can improve statistical efficiency and generalization, a theme surveyed broadly in geometric deep learning (Bronstein et al., 2021). Related progress includes $E(n)$-equivariant graph networks for physical reasoning (Satorras et al., 2021). Nevertheless, *vertex-wise conjugation gauge symmetry* differs qualitatively from global group actions: it is naturally expressed not as a single group acting on the whole input, but as a *groupoid-level* redundancy where each object (vertex) carries its own local symmetry.

In parallel, graph neural networks (GNNs) and message passing methods provide strong generic machinery for relational learning (Gilmer et al., 2017; Kipf & Welling, 2017; Xu et al., 2019; Veličković et al., 2018). However, in gauge-randomized settings these architectures are not invariant by construction: the matrix entries of edge transports can be scrambled by independently chosen vertex frames. Similarly, spectral summaries, which are the backbone of classical graph signal processing

and spectral graph learning (Shuman et al., 2013; Bruna et al., 2013; Defferrard et al., 2016), extend to *connection Laplacians* and related tools in synchronization and vector diffusion (Singer & Wu, 2012; Bandeira et al., 2012). While the resulting spectra are gauge-invariant, they can be a severe information bottleneck for tasks whose signal is carried by specific global holonomies.

Recent work on *sheaves, bundles, and higher-order topology in learning* offers additional structure. Sheaf neural networks and their spectral theory (Hansen & Gebhart, 2020; Hansen & Ghrist, 2019) as well as neural sheaf diffusion and its variants (Bodnar et al., 2022; Barbero et al., 2022; Zaghen et al., 2024) and bundle neural networks (Bamberger et al., 2024) highlight the value of transporting features along edges while enforcing consistency constraints. At the same time, topological deep learning and cell/simplicial complex networks stress that *2-cell structure* (faces) can be as important as the 1-skeleton for representing and reasoning about data (Hajij et al., 2022; Ebli et al., 2020; Hajij et al., 2020; Lim, 2020).

Our starting point is that a discrete $SO(3)$ edge connection on a cell complex $X$ induces holonomy functorially on edge-paths in the 1-skeleton $X^{(1)}$. Equivalently, it defines a functor

$$F_U : \mathcal{P}(X^{(1)}) \to \mathbf{B}(SO(3)),$$

where $\mathcal{P}(X^{(1)})$ denotes the path groupoid of the 1-skeleton and $\mathbf{B}(SO(3))$ is the one-object groupoid with endomorphism group $SO(3)$. Gauge transformations act as *natural isomorphisms* between such functors. In the flat/noiseless regime, namely when the holonomy around each face boundary is trivial, this path-holonomy functor descends through the 2-cell relations to a functor

$$\overline{F}_U : \Pi_1(X) \to \mathbf{B}(SO(3)),$$

where $\Pi_1(X)$ is the fundamental groupoid of the full cell complex. From this viewpoint, gauge-invariant learning should factor through functorial constructions and categorical invariants. In particular, loop holonomies are endomorphisms in $\mathbf{B}(SO(3))$, and applying trace/character-type functionals yields canonical conjugacy-invariant observables; this is reminiscent of categorical trace ideas (Joyal et al., 1996) and of compositional views of learning in categorical semantics (Fong et al., 2017).

We propose a new architecture for gauge-randomized $SO(3)$ learning problems, called Categorical Trace Loop Networks (CTLN). CTLN explicitly computes (i) holonomies of a fixed family of non-contractible torus generator loops and (ii) holonomies of plaquette (face-boundary) loops, and maps each holonomy to a conjugacy-invariant scalar via the rotation angle extracted from the trace. The resulting feature map is gauge-invariant by construction, while the inclusion of plaquette features injects 2-cell (sheaf/curvature) information that is diagnostically useful under noise.

The contributions of this work can be summarized as follows.

- We formulate a gauge-randomized $SO(3)$ torus connection regression benchmark that isolates global-topological signal under vertex-wise conjugation symmetry.
- We give CTLN, a novel model that combines groupoid composition (loop holonomy) with 2-cell (plaquette) structure and categorical trace/character scalarization.
- We provide a theoretical account: (i) gauge invariance of CTLN features, (ii) stability under multiplicative noise, and (iii) identifiability of global torus holonomy parameters.
- We empirically show in experiments that CTLN yields substantial improvements over MPNN (message passing neural network) and SpectralBundle (gauge-invariant spectral summary of a bundle/connection Laplacian).

The organization of the paper is as follows. Section 2 defines the torus connection model, gauge action, and task. Section 3 introduces CTLN. Section 4 provides invariance, stability, and identifiability results. Section 5 gives experimental protocol and results against MPNN and SpectralBundle.

## 2 PROBLEM SETUP: GAUGE-RANDOMIZED TORUS CONNECTIONS

### 2.1 TORUS CELL COMPLEX AND $SO(3)$ EDGE CONNECTION

Fix an integer $m \geq 2$ and write $\mathbb{Z}_m := \mathbb{Z}/m\mathbb{Z}$. We model the discrete torus $\mathbb{T}^2$ as the cubical cell complex $X_m = (V, E, F)$ with

$$V := \mathbb{Z}_m \times \mathbb{Z}_m.$$

We use the canonical generating directions $e_x = (1, 0)$ and $e_y = (0, 1)$ (arithmetic mod $m$). Define *positively oriented* horizontal and vertical edges

$$E_x := \{ (i, j) \to (i + 1, j) : (i, j) \in V \}, \qquad E_y := \{ (i, j) \to (i, j + 1) : (i, j) \in V \},$$

and set $E := E_x \sqcup E_y$. For each $e \in E$ we denote its source and target by $s(e), t(e) \in V$. We also consider *formal inverses*: for each $e \in E$ we introduce a reversed edge $\bar{e}$ with $s(\bar{e}) = t(e)$ and $t(\bar{e}) = s(e)$, and we write $\overline{E} := \{\bar{e} : e \in E\}$.

A (discrete) $\mathrm{SO}(3)$ *edge connection* on $X_m$ is an assignment

$$U : E \to \mathrm{SO}(3), \qquad e \mapsto U_e,$$

extended to reversed edges by the consistency rule

$$U_{\bar{e}} := U_e^{-1}.$$

Given a composable edge-path $p = (e_1, \ldots, e_L)$ (i.e. $t(e_\ell) = s(e_{\ell+1})$), its *holonomy* is the path-ordered product

$$H_U(p) := U_{e_L} \cdots U_{e_2} U_{e_1} \in \mathrm{SO}(3). \tag{1}$$

(We follow the convention that the first edge $e_1$ acts first.)

**Faces.**   Each face $f_{i,j} \in F$ (for $(i, j) \in V$) is the unit square with boundary loop

$$\partial f_{i,j} : \quad (i, j) \to (i + 1, j) \to (i + 1, j + 1) \to (i, j + 1) \to (i, j).$$

Let $e_r, e_u, e_l, e_d \in E$ denote respectively the *positively oriented* right, up, left, and down edges appearing in this square (so that the left and down traversals use inverses of $e_l$ and $e_d$). Then the *plaquette holonomy* (discrete curvature) is

$$H_U(\partial f_{i,j}) := U_{e_d}^{-1} U_{e_l}^{-1} U_{e_u} U_{e_r} \in \mathrm{SO}(3). \tag{2}$$

Each face-boundary loop $\partial f_{i,j}$ is contractible in the full cell complex $X_m$. For a flat connection on $X_m$, one has $H_U(\partial f_{i,j}) = I$ for every face $f_{i,j}$; departures from the identity quantify local inconsistency (discrete curvature) introduced by noise/outliers.

## 2.2   Gauge action and targets

**Gauge group and action.**   Let the vertex-wise gauge group be

$$\mathcal{G} := \mathrm{SO}(3)^V$$

with elements $g = (g_v)_{v \in V}$. The (right) gauge action on connections is

$$(g \cdot U)_e := g_{t(e)} U_e g_{s(e)}^{-1}, \qquad e \in E, \tag{3}$$

extended to $\bar{e}$ by inversion. Under this action, holonomy transforms by conjugation: for any closed loop/path $p$ based at $v$,

$$H_{g \cdot U}(p) = g_v H_U(p) g_v^{-1}. \tag{4}$$

**Data-generating process (gauge-randomized torus holonomies).**   Each sample is generated as follows.

1. Sample a commuting pair $(A, B) \in \mathrm{SO}(3) \times \mathrm{SO}(3)$ of global torus holonomies. In our experiments we restrict to a commuting family by choosing a random unit axis $u \in S^2$ and angles

   $$\alpha, \beta \in [0, \pi], \qquad A = \mathrm{Rot}(u, \alpha), \ B = \mathrm{Rot}(u, \beta),$$

   which ensures $AB = BA$.

2. Sample i.i.d. vertex gauges $g_v \sim \mathrm{Haar}(\mathrm{SO}(3))$ (i.e., uniformly at random from $\mathrm{SO}(3)$ wrt. the normalized Haar measure) for all $v \in V$.

3. Define a noiseless connection $U$ on edges by inserting $A$ and $B$ only on wrap-around edges:

$$U_{(i,j) \to (i+1,j)} := g_{(i+1,j)} A^{\mathbf{1}[i=m-1]} g_{(i,j)}^{-1}, \tag{5}$$

$$U_{(i,j) \to (i,j+1)} := g_{(i,j+1)} B^{\mathbf{1}[j=m-1]} g_{(i,j)}^{-1}, \tag{6}$$

where $\mathbf{1}[\cdot]$ is the indicator and arithmetic is mod $m$. When $A$ and $B$ commute, this construction yields a globally flat on all face-boundary loops, i.e. $H_U(\partial f_{i,j}) = I$ for all faces.

4. Apply independent multiplicative noise on each edge:

$$\widetilde{U}_e := N_e U_e, \qquad N_e = \exp(\xi_e), \ \xi_e \in \mathfrak{so}(3), \tag{7}$$

where $\xi_e$ is drawn from an isotropic Gaussian in the Lie algebra (scaled by a noise parameter $\sigma$).

**Learning target.**   Given the observed noisy, gauge-randomized connection $\widetilde{U}$, the learning task is to predict the two global holonomy angles

$$y := (\alpha, \beta) \in [0, \pi]^2.$$

Because conjugation does not change the rotation angle in $\mathrm{SO}(3)$, $(\alpha, \beta)$ is a natural gauge-invariant target for this benchmark.

## 2.3   Evaluation metrics

For a predictor $f$ outputting $\widehat{y} = (\widehat{\alpha}, \widehat{\beta})$ from $\widetilde{U}$, we report the test mean absolute error (MAE) averaged over both coordinates:

$$\mathrm{MAE}(\widehat{y}, y) := \frac{1}{2}\Big(|\widehat{\alpha} - \alpha| + |\widehat{\beta} - \beta|\Big). \tag{8}$$

Given a test set $\{(\widetilde{U}^{(n)}, y^{(n)})\}_{n=1}^N$, the reported test MAE is

$$\frac{1}{N} \sum_{n=1}^N \mathrm{MAE}\Big(f(\widetilde{U}^{(n)}), y^{(n)}\Big).$$

We train with fixed data splits and report *mean $\pm$ standard deviation* over three independent random initialization seeds, holding the data splits fixed across seeds.

# 3   Method: Categorical Trace Loop Network (CTLN)

## 3.1   Categorical invariant features from loops and faces

Let $X_m^{(1)}$ denote the 1-skeleton of $X_m$, and let $\mathcal{P}(X_m^{(1)})$ denote its path groupoid (objects are vertices; morphisms are edge-paths, modulo cancellation of immediate reversals). An $\mathrm{SO}(3)$ edge connection induces a functor

$$\mathcal{F}_U : \mathcal{P}(X_m^{(1)}) \to \mathbf{B}(\mathrm{SO}(3)),$$

where $\mathbf{B}(\mathrm{SO}(3))$ is the one-object groupoid with endomorphism group $\mathrm{SO}(3)$, by sending each edge-path to its holonomy equation 1. A gauge transformation $g \in \mathcal{G}$ acts as a *natural isomorphism* between $\mathcal{F}_U$ and $\mathcal{F}_{g \cdot U}$; concretely, this is the pathwise form of equation 4, and on loops it reduces to conjugation of holonomies at the base vertex. In the flat/noiseless regime, namely when the holonomy around every face boundary is trivial, the path-holonomy functor descends through the 2-cell relations to a functor

$$\overline{\mathcal{F}}_U : \Pi_1(X_m) \to \mathbf{B}(\mathrm{SO}(3)),$$

where $\Pi_1(X_m)$ is the fundamental groupoid of the full cell complex $X_m$.

**Loop families on the torus.**    We extract gauge-invariant information by evaluating $\mathcal{F}_{\widetilde{U}}$ on a fixed family of based loops in the 1-skeleton:

- **Row (non-contractible) loops.** For each $j \in \mathbb{Z}_m$, define the horizontal loop

$$\gamma_j : (0, j) \to (1, j) \to \cdots \to (m-1, j) \to (0, j).$$

- **Column (non-contractible) loops.** For each $i \in \mathbb{Z}_m$, define the vertical loop

$$\eta_i : (i, 0) \to (i, 1) \to \cdots \to (i, m-1) \to (i, 0).$$

- **Face-boundary loops (contractible in $X_m$).** For each $(i, j) \in \mathbb{Z}_m \times \mathbb{Z}_m$, let $\partial f_{i,j}$ denote the boundary loop of the plaquette $f_{i,j}$.

The row and column loops probe the two canonical non-contractible directions of the torus; in the flat/noiseless regime, they descend to representatives of the two generators of $\pi_1(T^2) \cong \mathbb{Z}^2$. The face-boundary loops probe local flatness/curvature consistency: they are contractible in the full cell complex $X_m$ (though not in the 1-skeleton viewed purely as a graph), and for a flat connection one has $H_U(\partial f_{i,j}) = I$ for every face.

**Conjugacy-invariant scalarization via trace/rotation angle.**    For any loop $\ell$ we compute its holonomy matrix

$$H(\ell) := H_{\widetilde{U}}(\ell) \in SO(3).$$

Under gauge transformations, $H(\ell)$ changes by conjugation, hence its conjugacy class is invariant. In $SO(3)$, the conjugacy class is determined by the *rotation angle* $\theta \in [0, \pi]$. We therefore map each holonomy to the scalar

$$\theta(H) := \arccos\left(\text{clamp}\left(\frac{\text{tr}(H) - 1}{2}, -1, 1\right)\right) \in [0, \pi], \tag{9}$$

where $\text{clamp}$ is included for numerical stability. Because $\text{tr}(gHg^{-1}) = \text{tr}(H)$, $\theta(H(\ell))$ is gauge-invariant for every loop $\ell$.

**CTLN feature map.**    Define the CTLN feature vector

$$\Phi(\widetilde{U}) := \begin{pmatrix} \underbrace{\theta(H(\gamma_0)), \ldots, \theta(H(\gamma_{m-1}))}_{m \text{ row loops}}, \\ \underbrace{\theta(H(\eta_0)), \ldots, \theta(H(\eta_{m-1}))}_{m \text{ column loops}}, \\ \underbrace{\theta(H(\partial f_{0,0})), \ldots, \theta(H(\partial f_{m-1,m-1}))}_{m^2 \text{ faces}} \end{pmatrix} \in [0, \pi]^{2m+m^2}. \tag{10}$$

By construction, $\Phi$ depends on $\widetilde{U}$ only through functorial composition (loop products) and conjugacy-invariant trace/angle statistics; it is therefore *invariant under the gauge action* equation 3.

### 3.2 PREDICTOR

CTLN predicts the global holonomy angles via a lightweight regressor on $\Phi(\widetilde{U})$:

$$(\widehat{\alpha}, \widehat{\beta}) = f_\theta(\Phi(\widetilde{U})), \qquad f_\theta : \mathbb{R}^{2m+m^2} \to \mathbb{R}^2, \tag{11}$$

where $f_\theta$ is a multilayer perceptron (MLP). Concretely (as in our implementation), we use two hidden layers with ReLU activations and a linear output layer. We train by minimizing mean squared error (MSE) over training samples,

$$\min_\theta \frac{1}{N} \sum_{n=1}^{N} \left\| f_\theta(\Phi(\widetilde{U}^{(n)})) - (\alpha^{(n)}, \beta^{(n)}) \right\|_2^2, \tag{12}$$

select the best checkpoint by validation MAE, and report test MAE as in equation 8.

## 4 MATHEMATICAL PROPERTIES OF CTLN

### 4.1 GAUGE INVARIANCE OF CTLN FEATURES

We formalize the gauge invariance of the CTLN feature map $\Phi$ in equation 10. Recall the vertex-wise gauge group $\mathcal{G} = \mathrm{SO}(3)^V$ acting on connections by equation 3. For a loop $\ell$ based at $v$, holonomy transforms by conjugation equation 4.

**Lemma 1** (Holonomy conjugation under gauge). *Let $U$ be a connection on $X_m$, let $g \in \mathcal{G}$, and let $\ell$ be a closed loop based at $v \in V$. Then*

$$H_{g \cdot U}(\ell) \;=\; g_v\, H_U(\ell)\, g_v^{-1}.$$

*Proof.* Write $\ell = (e_1, \ldots, e_L)$ as a composable edge sequence with $s(e_1) = t(e_L) = v$. By equation 3,

$$(g \cdot U)_{e_k} = g_{t(e_k)} U_{e_k} g_{s(e_k)}^{-1}.$$

Multiplying in path order and using $t(e_k) = s(e_{k+1})$ yields telescoping cancellation of intermediate $g$ factors, leaving $g_{t(e_L)}(\cdots)g_{s(e_1)}^{-1} = g_v(\cdots)g_v^{-1}$. $\qquad\square$

CTLN scalarizes holonomy matrices via the rotation-angle map $\theta : \mathrm{SO}(3) \to [0, \pi]$ defined in equation 9. This scalarization is conjugacy-invariant.

**Lemma 2** (Conjugacy invariance of $\theta$). *For any $R \in \mathrm{SO}(3)$ and any $Q \in \mathrm{SO}(3)$, $\theta(QRQ^{-1}) = \theta(R)$.*

*Proof.* Since $\mathrm{tr}(QRQ^{-1}) = \mathrm{tr}(R)$, the right-hand side of equation 9 is unchanged. $\qquad\square$

**Theorem 1** (Gauge invariance of CTLN features). *Let $\Phi$ be the CTLN feature map in equation 10. Then for every $g \in \mathcal{G}$ and every connection $U$ on $X_m$,*

$$\Phi(g \cdot U) \;=\; \Phi(U).$$

*Proof.* Each CTLN coordinate is of the form $\theta(H_U(\ell))$ for some loop $\ell$ (row loop, column loop, or face boundary loop). By Lemma 1, $H_{g \cdot U}(\ell) = g_v H_U(\ell) g_v^{-1}$ (for an appropriate base vertex $v$), hence by Lemma 2, $\theta(H_{g \cdot U}(\ell)) = \theta(H_U(\ell))$. Concatenating coordinates yields $\Phi(g \cdot U) = \Phi(U)$. $\qquad\square$

### 4.2 STABILITY UNDER MULTIPLICATIVE NOISE

We quantify how CTLN features change under multiplicative edge noise $\widetilde{U}_e = N_e U_e$ as in equation 7. We use the standard bi-invariant geodesic distance on $\mathrm{SO}(3)$:

$$d_{\mathrm{SO}(3)}(R, S) \;:=\; \theta(R^{-1}S) \in [0, \pi], \tag{13}$$

i.e. the rotation angle of the relative rotation $R^{-1}S$.

**Lemma 3** (Bi-invariance and product perturbation bound). *For all $A, A', B, B' \in \mathrm{SO}(3)$,*

$$d_{\mathrm{SO}(3)}(AB, A'B') \;\leq\; d_{\mathrm{SO}(3)}(A, A') \;+\; d_{\mathrm{SO}(3)}(B, B').$$

*Proof.* By the triangle inequality and right/left invariance of $d_{\mathrm{SO}(3)}$. $\qquad\square$

**Lemma 4** ($\theta$ is 1-Lipschitz w.r.t. $d_{\mathrm{SO}(3)}$). *For all $R, S \in \mathrm{SO}(3)$,*

$$|\theta(R) - \theta(S)| \;\leq\; d_{\mathrm{SO}(3)}(R, S).$$

*Proof.* In any metric space $(X, d)$, the map $x \mapsto d(x, z)$ is 1-Lipschitz for fixed $z$ since $|d(x, z) - d(y, z)| \leq d(x, y)$ by the triangle inequality. Here $X = \mathrm{SO}(3)$, $z = I$, and $\theta(R) = d_{\mathrm{SO}(3)}(R, I)$ by equation 13. $\qquad\square$

We now propagate edge-wise noise to loop-holonomy feature perturbations.

**Theorem 2** (Stability of loop angles under multiplicative edge noise). *Let $p = (e_1, \ldots, e_L)$ be any (not necessarily closed) path in the 1-skeleton, and let $\widetilde{U}_e = N_e U_e$ for $N_e \in \mathrm{SO}(3)$. Then*

$$d_{\mathrm{SO}(3)}\big(H_{\widetilde{U}}(p),\, H_U(p)\big) \;\leq\; \sum_{k=1}^{L} d_{\mathrm{SO}(3)}(N_{e_k}, I). \tag{14}$$

*Consequently, for any loop $\ell$,*

$$\big|\theta(H_{\widetilde{U}}(\ell)) - \theta(H_U(\ell))\big| \;\leq\; \sum_{e \in \ell} d_{\mathrm{SO}(3)}(N_e, I). \tag{15}$$

*Proof.* Write $H_{\widetilde{U}}(p) = \prod_{k=L}^{1}(N_{e_k} U_{e_k})$ and $H_U(p) = \prod_{k=L}^{1} U_{e_k}$. Applying Lemma 3 recursively yields

$$d\left( \prod_{k=L}^{1} N_{e_k} U_{e_k},\, \prod_{k=L}^{1} U_{e_k} \right) \leq \sum_{k=1}^{L} d(N_{e_k} U_{e_k}, U_{e_k}).$$

By right invariance, $d(N_{e_k} U_{e_k}, U_{e_k}) = d(N_{e_k}, I)$, proving equation 14. Then equation 15 follows from Lemma 4 with $R = H_{\widetilde{U}}(\ell)$ and $S = H_U(\ell)$. □

Since $\Phi(\widetilde{U})$ is a concatenation of loop-angle maps $\theta(H_{\widetilde{U}}(\ell))$ over rows, columns, and faces, Theorem 2 gives an explicit robustness bound for each coordinate. In particular, each plaquette feature depends on a length-4 loop and is therefore *locally* stable, while each torus-generator feature depends on a length-$m$ loop, scaling linearly with the path length under worst-case perturbations.

### 4.3  IDENTIFIABILITY ON THE TORUS AND COMMUTING HOLONOMIES

We show that in the noiseless generative model of Section 2, the target angles $(\alpha, \beta)$ are identifiable from CTLN loop invariants. The key structural property is flatness on contractible faces, which on the torus implies a representation of $\pi_1(\mathbb{T}^2) \cong \mathbb{Z}^2$ by a *commuting* pair of holonomies.

**Proposition 1** (Noiseless CTLN features recover $(\alpha, \beta)$). *Consider the noiseless construction in Section 2.2 with $\widetilde{U} = U$ (i.e. $N_e = I$ for all edges), where $A = \mathrm{Rot}(u, \alpha)$ and $B = \mathrm{Rot}(u, \beta)$. Then for every row loop $\gamma_j$ and every column loop $\eta_i$,*

$$\theta(H_U(\gamma_j)) = \alpha, \qquad \theta(H_U(\eta_i)) = \beta,$$

*and for every face $f_{i,j}$,*

$$\theta(H_U(\partial f_{i,j})) = 0.$$

*Consequently, $(\alpha, \beta)$ is identifiable from $\Phi(U)$ (indeed, it appears explicitly as coordinates).*

*Proof.* Fix a row loop $\gamma_j$ based at $(0, j)$. Along $\gamma_j$, every non-wrap horizontal edge has $U_{(i,j)\to(i+1,j)} = g_{(i+1,j)} g_{(i,j)}^{-1}$, while the unique wrap edge $(m-1, j) \to (0, j)$ has $U_{(m-1,j)\to(0,j)} = g_{(0,j)} A g_{(m-1,j)}^{-1}$. Multiplying holonomies around the loop telescopes all intermediate gauges, yielding

$$H_U(\gamma_j) = g_{(0,j)} A g_{(0,j)}^{-1}.$$

Thus $\theta(H_U(\gamma_j)) = \theta(A) = \alpha$ by conjugacy invariance (Lemma 2). The column case is identical with $B$.

For a face loop $\partial f_{i,j}$ not crossing any wrap boundary, the holonomy is the identity by the same telescoping cancellation. For faces adjacent to wrap boundaries, the plaquette holonomy is a conjugate of the commutator $[A, B] = A^{-1} B^{-1} A B$. Since in our generative model $A$ and $B$ commute (because they share the same axis), $[A, B] = I$ and hence $H_U(\partial f_{i,j}) = I$ for all faces. Therefore $\theta(H_U(\partial f_{i,j})) = 0$ for all $(i, j)$. □

Proposition 1 shows that (in the idealized noiseless case) row/column loop angles already identify $(\alpha, \beta)$. In the noisy regime, however, different corruption patterns can produce similar row/column loop angles while inducing markedly different local curvature patterns. The plaquette angles $\theta(H_{\widetilde{U}}(\partial f_{i,j}))$ thus provide an additional, *local* consistency signal that enables the predictor to adapt its interpretation of global loop measurements, e.g. by down-weighting regimes where local inconsistencies indicate heavy edge corruption.

## 5  EXPERIMENTS AND RESULTS

We compare CTLN against two baselines.

**MPNN.**  The MPNN baseline is a standard message passing neural network on the 1-skeleton of $X_m$. We represent each (directed) edge by a feature vector consisting of the 9 matrix entries of $\widetilde{U}_e \in \mathrm{SO}(3)$ and an edge-type indicator (horizontal/vertical). Node states are initialized as learned constants and updated for $L$ rounds by aggregating incoming messages computed by an MLP. A global mean-pooling readout over node states produces the prediction $(\widehat{\alpha}, \widehat{\beta})$. Importantly, this model is *not* gauge-equivariant/invariant by construction under the vertex-wise conjugation action equation 3.

**SpectralBundle.**  SpectralBundle computes a gauge-invariant spectral summary of a bundle/connection Laplacian built from $\widetilde{U}$. Let $|V| = m^2$ and identify $\mathbb{R}^{3|V|} \cong \bigoplus_{v \in V} \mathbb{R}^3$. Define the block matrix $L(\widetilde{U}) \in \mathbb{R}^{3|V| \times 3|V|}$ by

$$L_{vv} = \deg(v)\, I_3, \qquad L_{vw} = \begin{cases} -\widetilde{U}_{v \to w} & \text{if } (v \to w) \in E, \\ -\widetilde{U}_{w \to v}^{\top} & \text{if } (w \to v) \in E, \\ 0 & \text{otherwise,} \end{cases}$$

where $\deg(v) = 4$ on the torus. Under a gauge transform $g = (g_v)_{v \in V}$, the matrix transforms as $L(g \cdot \widetilde{U}) = G\, L(\widetilde{U})\, G^{-1}$ with $G = \mathrm{diag}(g_v)_{v \in V}$, hence the spectrum is gauge-invariant. SpectralBundle takes the smallest $k$ eigenvalues of $L(\widetilde{U})$ (sorted) as features and feeds them to a 2-hidden-layer MLP to regress $(\alpha, \beta)$.

### 5.1  EXPERIMENTAL PROTOCOL

**Data.**  We use the gauge-randomized $\mathrm{SO}(3)$ torus generator in Section 2.2 with $m = 4$. Each sample has $|E| = 32$ edge matrices in $\mathrm{SO}(3)$. We draw $\alpha, \beta \sim \mathrm{Unif}[0, \pi]$ and a random axis $u \sim \mathrm{Unif}(S^2)$, and set $A = \mathrm{Rot}(u, \alpha)$, $B = \mathrm{Rot}(u, \beta)$. Vertex gauges are sampled i.i.d. from the Haar measure on $\mathrm{SO}(3)$. Edge noise is multiplicative, $\widetilde{U}_e = N_e U_e$, where $N_e = \exp(\xi_e)$ and $\xi_e \sim \mathcal{N}(0, \sigma^2 I)$ in $\mathfrak{so}(3)$ with $\sigma = 0.20$.

**Splits and seeds.**  We use fixed splits of size $N_{\text{train}} = 1500$, $N_{\text{val}} = 300$, and $N_{\text{test}} = 300$, and report mean $\pm$ standard deviation over three random initialization seeds (seeds $0, 1, 2$), keeping the data splits fixed across seeds.

**Training.**  All learned models (CTLN, MPNN, SpectralBundle) are trained to minimize MSE on $(\alpha, \beta)$ with Adam (learning rate $10^{-3}$), ReLU activations, and a fixed training budget of 15 epochs. We select the checkpoint with the best validation MAE and report test MAE computed as in equation 8.

### 5.2  RESULTS

Table 1 reports the main results. CTLN achieves a test MAE of $0.1747 \pm 0.0018$, significantly outperforming both MPNN and SpectralBundle on this gauge-randomized benchmark.[1]

We can analyze the results as follows. The gauge action equation 3 is vertex-wise conjugation, i.e. a *local* (groupoid) symmetry. Under random vertex gauges, the raw matrix entries of $\widetilde{U}_e$ vary by conjugation that depends on the endpoints of each edge. A generic MPNN operating on matrix entries must therefore implicitly learn a nonlinear gauge-fixing/quotient operation to extract signal about $(\alpha, \beta)$. Absent an architectural constraint enforcing gauge-invariance/equivariance, the learning problem becomes highly ill-conditioned: the target depends on global holonomies (non-contractible loops), while local edge features are dominated by arbitrary gauge.

---

[1] To rule out undertraining, we also ran 50-epoch training with early stopping (patience 10), which did not materially improve MPNN/SpectralBundle.

Table 1: Test MAE (mean $\pm$ std over 3 seeds). Lower is better.

| Model | Test MAE $\downarrow$ |
|---|---|
| Ours: CTLN (LoopMLP+Faces) | $0.1747 \pm 0.0018$ |
| SpectralBundle | $0.8105 \pm 0.0009$ |
| MPNN | $0.8134 \pm 0.0020$ |

SpectralBundle is gauge-invariant by design (spectrum of a conjugated operator is unchanged), but the spectral summary is an aggressive information bottleneck: the map from global holonomy parameters $(\alpha, \beta)$ to the finite spectral prefix can be weakly informative and non-injective, especially under multiplicative noise. In contrast, CTLN directly measures Wilson-loop-type observables of the two torus generators via functorial loop composition, producing high signal-to-noise features for $(\alpha, \beta)$ (Proposition 1) and stable perturbation behavior (Theorem 2).

Row/column loop angles probe global topology, while plaquette angles probe local curvature/inconsistency. Under edge noise, curvature features provide additional diagnostic information about where local constraints are violated, enabling the predictor to disambiguate noisy global loop measurements. This is precisely the sense in which CTLN exploits *both* groupoid composition (1-cells) and sheaf/curvature structure (2-cells) in a single categorical representation.

## 6  CONCLUSION

We developed Categorical Trace Loop Networks (CTLN), which combine global loop observables with local 2-cell consistency signals, yielding a compact gauge-invariant representation that preserves the information relevant to global holonomy. Empirically, CTLN produces a large and consistent improvement over both message passing and spectral connection-Laplacian summaries on gauge-randomized $SO(3)$ torus holonomy regression. Beyond the benchmark, the significance is conceptual: categorical constructions can provide the right interface between symmetry, topology, and learning, particularly when the signal lives in global structure rather than in local coordinates.

The present work can be viewed as a first step from *categorical invariants* toward *category-equivariant learning*. In the current model, the categorical structure enters through the representation of a connection as a functor $\mathcal{F}_U : \mathcal{P}(X_m^{(1)}) \to \mathbf{B}(SO(3))$, together with gauge transformations acting as natural isomorphisms, and the network ultimately reads out gauge-invariant quantities derived from selected loop holonomies. A category-equivariant extension would retain the functorial viewpoint throughout the network: hidden states would themselves be assigned functorially to objects, paths, cells, or local systems; layer maps would be required to commute with the morphisms of the underlying category; and gauge transformations would act equivariantly in the latent space, with invariance imposed only at the final readout.

Concrete research directions follow from this perspective. First, one can replace the fixed loop feature extractor by a category-equivariant message-passing scheme on path-, groupoid-, or cell-level representations, allowing the model to learn which morphism compositions are task-relevant rather than pre-specifying row, column, and plaquette loops. Second, one can extend from the one-object target groupoid $\mathbf{B}(SO(3))$ to richer target categories encoding associated representations, local coefficient systems, or sheaf-valued features, thereby combining holonomy, curvature, and higher-order consistency in a single equivariant architecture.

More broadly, the results here support the view that topology, gauge structure, and categorical composition are not merely descriptive language for geometric learning problems, but can determine what information is statistically accessible in the first place. In the benchmark studied here, the relevant signal is global and functorial rather than local and coordinate-based; this is exactly the regime where categorical constructions become operationally useful. We therefore see CTLN not as the end of the story, but as evidence that category-aware inductive biases can be turned into effective learning principles, and as a foundation for future fully category-equivariant architectures that operate directly on functorial and higher-categorical representations.[2]

---

[2]We refer to Maruyama (2025a;b;c); Xu & Maruyama (2022); Sennesh et al. (2022; 2023); Abbott et al. (2024; 2026); Maruyama & Yasuda (2025a;b); Maruyama (2026a;b) for our related categorical ML work.

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

## A  Mathematical Details and Additional Results

In this appendix, we provide full details for the mathematical results above as well as several additional results. Let $X_m = (V, E, F)$ denote the $m \times m$ torus cell complex from Section 2.1. We write $\mathrm{SO}(3)$ for the rotation group and use the conventions of equation 1, equation 3, equation 2, and equation 9.

### A.1  Basic facts about $\mathrm{SO}(3)$ and the trace–angle map

We first justify that the CTLN scalarization $\theta(\cdot)$ is well-defined and conjugacy-invariant.

**Lemma 5** (Rotation angle and trace identity). *For every $R \in \mathrm{SO}(3)$ there exists a unit vector $u \in S^2$ and a unique angle $\theta \in [0, \pi]$ such that $R = \mathrm{Rot}(u, \theta)$ (rotation by $\theta$ around axis $u$). Moreover,*

$$\mathrm{tr}(R) \;=\; 1 + 2\cos\theta, \qquad \textit{hence} \qquad \theta \;=\; \arccos\!\left(\frac{\mathrm{tr}(R) - 1}{2}\right) \in [0, \pi]. \tag{16}$$

*Proof.* Existence of an axis-angle representation with $\theta \in [0, \pi]$ is classical: any $R \in \mathrm{SO}(3)$ has an eigenvector $u$ with eigenvalue 1 (because $\det(R - I) = 0$ for $R \in \mathrm{SO}(3)$ unless $R = I$, and in all cases the characteristic polynomial has a real root). The action of $R$ on the orthogonal complement $u^\perp \cong \mathbb{R}^2$ is a planar rotation by some angle $\theta \in [0, 2\pi)$; choosing $\theta \in [0, \pi]$ yields uniqueness.

For the trace identity, note that the eigenvalues of $R$ are $\{1, e^{i\theta}, e^{-i\theta}\}$ for the same $\theta \in [0, \pi]$ (the complex conjugate pair corresponds to the planar rotation on $u^\perp$). Therefore $\mathrm{tr}(R) = 1 + e^{i\theta} + e^{-i\theta} = 1 + 2\cos\theta$, and rearranging gives equation 16. $\qquad\square$

**Lemma 6** (Conjugacy invariance of $\theta$). *For any $R, Q \in \mathrm{SO}(3)$,*

$$\theta(QRQ^{-1}) = \theta(R).$$

*Proof.* $\mathrm{tr}(QRQ^{-1}) = \mathrm{tr}(R)$, hence the right-hand side of equation 9 (and equation 16) is unchanged. $\qquad\square$

**Numerical clamp.** In practice we implement equation 9 with a clamped argument to $\arccos$ to avoid small floating-point violations of $[-1, 1]$. This does not affect the mathematical statements below, which assume exact arithmetic.

### A.2  A bi-invariant metric on $\mathrm{SO}(3)$

Define

$$d_{\mathrm{SO}(3)}(R, S) \;:=\; \theta(R^{-1}S) \in [0, \pi]. \tag{17}$$

This is the standard geodesic distance induced by the bi-invariant Riemannian metric on $\mathrm{SO}(3)$. We include a proof of the metric properties, since we use the triangle inequality in stability arguments.

**Lemma 7** (Quaternion expression for $d_{\mathrm{SO}(3)}$). *Let $q_1, q_2 \in S^3 \subset \mathbb{R}^4$ be unit quaternions representing $R, S \in \mathrm{SO}(3)$ under the standard double cover $S^3 \to \mathrm{SO}(3)$ (so $q$ and $-q$ represent the same rotation). Then*

$$d_{\mathrm{SO}(3)}(R, S) = 2 \arccos(|\langle q_1, q_2 \rangle|), \qquad (18)$$

*where $\langle \cdot, \cdot \rangle$ is the Euclidean inner product on $\mathbb{R}^4$.*

*Proof.* Let $q_R, q_S$ be unit quaternions representing $R, S$. The relative rotation $R^{-1}S$ is represented by the quaternion $q := q_R^{-1} q_S$ (quaternion multiplication), and $q$ is unit. Write $q = (q_0, \vec{q})$ with scalar part $q_0 \in \mathbb{R}$ and vector part $\vec{q} \in \mathbb{R}^3$. The axis-angle representation associated to $q$ is rotation by angle $2 \arccos(|q_0|)$ around axis $\vec{q}/\|\vec{q}\|$. Thus $\theta(R^{-1}S) = 2 \arccos(|q_0|)$.

It remains to identify $|q_0|$ with $|\langle q_R, q_S \rangle|$. For unit quaternions, $q_R^{-1} = \bar{q}_R = (q_{R,0}, -q_{R,1}, -q_{R,2}, -q_{R,3})$, and the scalar part of $\bar{q}_R q_S$ equals the Euclidean inner product $\langle q_R, q_S \rangle$. Taking absolute value removes the sign ambiguity ($q_R$ and $-q_R$ represent the same rotation), yielding equation 18. $\qquad\square$

**Theorem 3** ($d_{\mathrm{SO}(3)}$ is a bi-invariant metric). *The function $d_{\mathrm{SO}(3)}$ in equation 17 is a metric on $\mathrm{SO}(3)$ and is bi-invariant: for all $R, S, Q \in \mathrm{SO}(3)$,*

$$d_{\mathrm{SO}(3)}(QR, QS) = d_{\mathrm{SO}(3)}(R, S), \qquad d_{\mathrm{SO}(3)}(RQ, SQ) = d_{\mathrm{SO}(3)}(R, S).$$

*Proof. Nonnegativity and symmetry* follow from $\theta(\cdot) \in [0, \pi]$ and $\theta(T^{-1}) = \theta(T)$. Indeed, $d(R, S) = \theta(R^{-1}S) = \theta((R^{-1}S)^{-1}) = \theta(S^{-1}R) = d(S, R)$.

*Identity of indiscernibles.* If $d(R, S) = 0$, then $\theta(R^{-1}S) = 0$, hence $R^{-1}S = I$ by Lemma 5, so $R = S$. Conversely, $d(R, R) = \theta(I) = 0$.

*Bi-invariance.* Left invariance is immediate:

$$d(QR, QS) = \theta((QR)^{-1}(QS)) = \theta(R^{-1}S) = d(R, S).$$

For right invariance,

$$d(RQ, SQ) = \theta((RQ)^{-1}(SQ)) = \theta(Q^{-1}R^{-1}SQ).$$

By Lemma 6, $\theta(Q^{-1}R^{-1}SQ) = \theta(R^{-1}S)$, hence $d(RQ, SQ) = d(R, S)$.

*Triangle inequality.* Let $R, S, T \in \mathrm{SO}(3)$ and choose unit quaternions $q_R, q_S, q_T \in S^3$ representing them. Define the spherical distance on $S^3$ by

$$\delta(q_1, q_2) := \arccos(\langle q_1, q_2 \rangle) \in [0, \pi],$$

which is the geodesic distance on the unit sphere and hence satisfies the triangle inequality. Consider the quotient by the isometric action $\{\pm 1\}$ on $S^3$ and define

$$\bar{\delta}([q_1], [q_2]) := \min\{\delta(q_1, q_2), \ \delta(q_1, -q_2)\}.$$

We claim $\bar{\delta}$ satisfies the triangle inequality. Fix $\varepsilon > 0$ and choose signs $s_{12}, s_{23} \in \{\pm 1\}$ such that

$$\bar{\delta}([q_R], [q_S]) \geq \delta(q_R, s_{12}q_S) - \varepsilon, \qquad \bar{\delta}([q_S], [q_T]) \geq \delta(q_S, s_{23}q_T) - \varepsilon.$$

Then, using the triangle inequality for $\delta$ and that sign flips are isometries,

$$\delta(q_R, s_{12}s_{23}q_T) \leq \delta(q_R, s_{12}q_S) + \delta(s_{12}q_S, s_{12}s_{23}q_T) = \delta(q_R, s_{12}q_S) + \delta(q_S, s_{23}q_T).$$

Taking the minimum over $\pm q_T$ on the left gives

$$\bar{\delta}([q_R], [q_T]) \leq \delta(q_R, s_{12}s_{23}q_T) \leq \bar{\delta}([q_R], [q_S]) + \bar{\delta}([q_S], [q_T]) + 2\varepsilon.$$

Letting $\varepsilon \to 0$ yields $\bar{\delta}([q_R], [q_T]) \leq \bar{\delta}([q_R], [q_S]) + \bar{\delta}([q_S], [q_T])$.

Finally, by Lemma 7, $d_{\mathrm{SO}(3)}(R, S) = 2\bar{\delta}([q_R], [q_S])$, so scaling preserves the triangle inequality. $\qquad\square$

**Lemma 8** (A convenient product perturbation inequality). *For all $A, A', B, B' \in \mathrm{SO}(3)$,*

$$d_{\mathrm{SO}(3)}(AB, A'B') \leq d_{\mathrm{SO}(3)}(A, A') + d_{\mathrm{SO}(3)}(B, B').$$

*Proof.* By triangle inequality and bi-invariance (Theorem 3),

$$d(AB, A'B') \leq d(AB, A'B) + d(A'B, A'B') = d(A, A') + d(B, B').$$

$\qquad\square$

### A.3   GAUGE INVARIANCE OF CTLN FEATURES

We restate and prove gauge invariance of the CTLN feature map.

**Lemma 9** (Holonomy conjugation under gauge). *Let $U$ be a connection, $g \in \mathcal{G} = \mathrm{SO}(3)^V$, and let $p = (e_1, \ldots, e_L)$ be a path with $s(e_1) = v$ and $t(e_L) = w$. Then*

$$H_{g \cdot U}(p) \;=\; g_w \, H_U(p) \, g_v^{-1}.$$

*In particular, if $p$ is a loop based at $v$ (so $w = v$), then $H_{g \cdot U}(p) = g_v H_U(p) g_v^{-1}$.*

*Proof.* By definition, $(g \cdot U)_{e_k} = g_{t(e_k)} U_{e_k} g_{s(e_k)}^{-1}$. Therefore

$$H_{g \cdot U}(p) = (g \cdot U)_{e_L} \cdots (g \cdot U)_{e_1} = \left( g_{t(e_L)} U_{e_L} g_{s(e_L)}^{-1} \right) \cdots \left( g_{t(e_1)} U_{e_1} g_{s(e_1)}^{-1} \right).$$

Because $s(e_{k+1}) = t(e_k)$, the adjacent factors $g_{s(e_{k+1})}^{-1} g_{t(e_k)}$ cancel, leaving

$$H_{g \cdot U}(p) = g_{t(e_L)} \, (U_{e_L} \cdots U_{e_1}) \, g_{s(e_1)}^{-1} = g_w \, H_U(p) \, g_v^{-1}.$$

$\square$

**Theorem 4** (Gauge invariance of CTLN features). *Let $\Phi$ be the CTLN feature map in equation 10, i.e. the concatenation of $\theta(H_U(\ell))$ over the chosen family of row loops, column loops, and face-boundary loops. Then $\Phi(g \cdot U) = \Phi(U)$ for all $g \in \mathcal{G}$.*

*Proof.* Each coordinate of $\Phi$ is of the form $\theta(H_U(\ell))$ for some loop $\ell$. By Lemma 9, $H_{g \cdot U}(\ell) = g_v H_U(\ell) g_v^{-1}$ for the base vertex $v$. By Lemma 6, $\theta(g_v H_U(\ell) g_v^{-1}) = \theta(H_U(\ell))$. Thus each coordinate is unchanged, hence $\Phi(g \cdot U) = \Phi(U)$. $\square$

### A.4   STABILITY UNDER MULTIPLICATIVE NOISE

Let $U$ be a connection and let $\widetilde{U}$ be the multiplicatively perturbed connection $\widetilde{U}_e = N_e U_e$ with $N_e \in \mathrm{SO}(3)$.

**Lemma 10** ($\theta$ is 1-Lipschitz w.r.t. $d_{\mathrm{SO}(3)}$). *For all $R, S \in \mathrm{SO}(3)$,*

$$|\theta(R) - \theta(S)| \;\leq\; d_{\mathrm{SO}(3)}(R, S).$$

*Proof.* By definition, $\theta(R) = d_{\mathrm{SO}(3)}(R, I)$ (see equation 17), hence

$$|\theta(R) - \theta(S)| = |d(R, I) - d(S, I)| \leq d(R, S)$$

by the triangle inequality for $d_{\mathrm{SO}(3)}$ (Theorem 3). $\square$

**Theorem 5** (Holonomy perturbation bound). *Let $p = (e_1, \ldots, e_L)$ be any path. If $\widetilde{U}_e = N_e U_e$, then*

$$d_{\mathrm{SO}(3)}\big(H_{\widetilde{U}}(p), H_U(p)\big) \;\leq\; \sum_{k=1}^{L} d_{\mathrm{SO}(3)}(N_{e_k}, I). \tag{19}$$

*Consequently, for any loop $\ell$,*

$$\big|\theta(H_{\widetilde{U}}(\ell)) - \theta(H_U(\ell))\big| \;\leq\; \sum_{e \in \ell} d_{\mathrm{SO}(3)}(N_e, I). \tag{20}$$

*Proof.* We prove equation 19 by induction on $L$. For $L = 1$, $H_{\widetilde{U}}(p) = N_{e_1} U_{e_1}$ and $H_U(p) = U_{e_1}$, so by right invariance of $d_{\mathrm{SO}(3)}$,

$$d(N_{e_1} U_{e_1}, U_{e_1}) = d(N_{e_1}, I),$$

which matches the claim.

Assume the claim holds for length $L - 1$. Write $p = (e_1, \ldots, e_{L-1}, e_L)$ and define

$$P := H_{\widetilde{U}}(e_1, \ldots, e_{L-1}), \quad Q := H_U(e_1, \ldots, e_{L-1}).$$

Then

$$H_{\widetilde{U}}(p) = \widetilde{U}_{e_L}\, P = (N_{e_L} U_{e_L})P, \qquad H_U(p) = U_{e_L}\, Q.$$

By Lemma 8,

$$d\big((N_{e_L} U_{e_L})P,\ U_{e_L}Q\big) \leq d(N_{e_L} U_{e_L}, U_{e_L}) + d(P,Q).$$

As before, $d(N_{e_L} U_{e_L}, U_{e_L}) = d(N_{e_L}, I)$ by right invariance, and $d(P,Q)$ is bounded by the induction hypothesis:

$$d(P,Q) \leq \sum_{k=1}^{L-1} d(N_{e_k}, I).$$

Summing yields equation 19.

For equation 20, apply Lemma 10 with $R = H_{\widetilde{U}}(\ell)$ and $S = H_U(\ell)$ and then use equation 19. $\qquad\square$

**CTLN feature stability.**  Since each CTLN coordinate is a loop-angle $\theta(H_{\widetilde{U}}(\ell))$, the bound equation 20 directly yields coordinate-wise robustness of $\Phi(\widetilde{U})$. In particular, plaquette features correspond to 4-edge loops and are therefore locally stable, while generator-loop features scale linearly with $m$ in the worst case.

## A.5   IDENTIFIABILITY ON THE TORUS IN THE COMMUTING-HOLONOMY REGIME

We prove that in the noiseless generative model, CTLN features determine the target angles.

**Proposition 2** (Noiseless recovery of $(\alpha, \beta)$ from CTLN features). *Assume the noiseless setting ($N_e = I$ for all edges) of Section 2.2 with $A = \mathrm{Rot}(u, \alpha)$ and $B = \mathrm{Rot}(u, \beta)$ where $\alpha, \beta \in [0, \pi]$. Let $U$ be the resulting connection. Then for every row loop $\gamma_j$ and column loop $\eta_i$,*

$$\theta(H_U(\gamma_j)) = \alpha, \qquad \theta(H_U(\eta_i)) = \beta,$$

*and for every face $f_{i,j}$,*

$$\theta(H_U(\partial f_{i,j})) = 0.$$

*In particular, $(\alpha, \beta)$ is identifiable from $\Phi(U)$.*

*Proof.* Fix a row index $j \in \mathbb{Z}_m$ and consider the row loop

$$\gamma_j : (0,j) \to (1,j) \to \cdots \to (m-1, j) \to (0, j).$$

By construction of $U$, for $i = 0, \ldots, m-2$ we have $U_{(i,j)\to(i+1,j)} = g_{(i+1,j)} g_{(i,j)}^{-1}$, while the wrap edge $(m-1,j) \to (0,j)$ has $U_{(m-1,j)\to(0,j)} = g_{(0,j)} A g_{(m-1,j)}^{-1}$. Therefore the holonomy along $\gamma_j$ is

$$\begin{aligned}
H_U(\gamma_j) &= U_{(m-1,j)\to(0,j)} \cdot U_{(m-2,j)\to(m-1,j)} \cdots U_{(0,j)\to(1,j)} \\
&= \big(g_{(0,j)} A g_{(m-1,j)}^{-1}\big) \cdot \big(g_{(m-1,j)} g_{(m-2,j)}^{-1}\big) \cdots \big(g_{(1,j)} g_{(0,j)}^{-1}\big) \\
&= g_{(0,j)}\, A\, g_{(0,j)}^{-1},
\end{aligned}$$

where all intermediate $g$ factors telescope. By conjugacy invariance (Lemma 6) and Lemma 5, $\theta(H_U(\gamma_j)) = \theta(A) = \alpha$.

The column case is identical with $B$ in place of $A$, yielding $\theta(H_U(\eta_i)) = \beta$.

For a face $f_{i,j}$, the plaquette holonomy is (by equation 2) $H_U(\partial f_{i,j}) = U_{e_d}^{-1} U_{e_l}^{-1} U_{e_u} U_{e_r}$. If the face does not cross a wrap boundary, each edge is of the pure-gauge form $g_{t(e)} g_{s(e)}^{-1}$, and a direct cancellation shows $H_U(\partial f_{i,j}) = I$. If the face crosses a wrap boundary, then $A$ and/or $B$ appear, and one obtains $H_U(\partial f_{i,j})$ as a conjugate of the commutator $A^{-1}B^{-1}AB$ (the discrete curvature of the global holonomies). Because our generative model enforces $AB = BA$, this commutator is $I$. Thus $H_U(\partial f_{i,j}) = I$ for all faces, so $\theta(H_U(\partial f_{i,j})) = 0$. $\qquad\square$

**Remark (why we assume a commuting family).**  On the torus, a flat connection corresponds to a representation of $\pi_1(\mathbb{T}^2) \cong \mathbb{Z}^2$ into $\mathrm{SO}(3)$, i.e. a commuting pair. Our benchmark enforces commutativity by construction by restricting to same-axis rotations; this yields a clean identifiability statement and isolates gauge effects.

A.6 Non-locality and a contractible-patch trivialization lemma

CTLN computes holonomies on *non-contractible* loops; we now formalize why purely local, gauge-invariant information cannot recover global torus holonomies in the flat/noiseless regime.

**Definition 1** (Flatness on a 2-complex). Let $Y = (V_Y, E_Y, F_Y)$ be a 2-dimensional cell complex (with oriented edges and faces). A connection $U : E_Y \to \mathrm{SO}(3)$ is *flat on $Y$* if for every face $f \in F_Y$, the holonomy around its boundary loop satisfies $H_U(\partial f) = I$.

**Lemma 11** (Flat connections are gauge-trivial on contractible complexes). *Let $Y$ be a connected, contractible 2-complex, and let $U$ be a flat connection on $Y$. Then there exists a gauge $g \in \mathrm{SO}(3)^{V_Y}$ such that $(g \cdot U)_e = I$ for every edge $e \in E_Y$.*

*Proof.* Fix a base vertex $v_0 \in V_Y$ and set $g_{v_0} := I$. For any vertex $v \in V_Y$, choose an edge-path $p_{v_0 \to v}$ from $v_0$ to $v$ in the 1-skeleton of $Y$ and define

$$g_v := H_U(p_{v_0 \to v}) \in \mathrm{SO}(3).$$

We claim this is well-defined (independent of the choice of path). Let $p$ and $p'$ be two paths from $v_0$ to $v$. Then $p \cdot \overline{p'}$ is a closed loop based at $v_0$. Because $Y$ is contractible, this loop is null-homotopic; in a 2-complex, null-homotopies are generated by inserting/removing face boundaries. Flatness (Definition 1) implies that inserting or removing a face boundary does not change holonomy (the inserted boundary contributes $I$). Therefore $H_U(p) = H_U(p')$, proving well-definedness.

Now consider any edge $e : u \to v$ in $Y$. Choose a path $p_{v_0 \to u}$ from $v_0$ to $u$ and extend it by $e$ to obtain a path $p_{v_0 \to v} = p_{v_0 \to u} \cdot e$ from $v_0$ to $v$. Then, by functoriality of holonomy,

$$g_v = H_U(p_{v_0 \to v}) = H_U(e) \, H_U(p_{v_0 \to u}) = U_e \, g_u.$$

Rearranging gives $U_e = g_v g_u^{-1}$, hence if we gauge-transform by $h_v := g_v^{-1}$, then

$$(h \cdot U)_e = h_v U_e h_u^{-1} = g_v^{-1}(g_v g_u^{-1})g_u = I.$$

$\square$

**Lemma 12** (Radius-$r$ patches are contractible when $r < m/2$). *Fix $m$ and let $r < m/2$. For any vertex $v \in V$ in the $m \times m$ torus complex $X_m$, the induced subcomplex $Y$ on vertices within graph distance $\leq r$ from $v$ is contractible and does not wrap around the torus.*

*Proof.* Lift coordinates from $\mathbb{Z}_m \times \mathbb{Z}_m$ to $\mathbb{Z}^2$ by choosing a representative $\tilde{v} \in \mathbb{Z}^2$ for $v$. Because $r < m/2$, any vertex at distance $\leq r$ from $v$ has a unique lift within the box $\tilde{v} + [-r, r]^2$ (no wrap-around ambiguity occurs). Thus the radius-$r$ neighborhood embeds in the planar grid as a finite rectangular patch with its faces, which is contractible. $\square$

**Theorem 6** (Local gauge-invariant information cannot recover global torus holonomies). *Consider the noiseless benchmark family from Section 2.2 obtained by varying the global holonomies $(A, B)$ (equivalently $(\alpha, \beta)$) while sampling arbitrary vertex gauges. Fix $r < m/2$. Let $Y$ be any radius-$r$ patch subcomplex of $X_m$ and let $U^{A,B}$ denote the noiseless connection generated from $(A, B)$. Then the restrictions $U^{A,B}|_Y$ are all gauge-equivalent to the trivial connection on $Y$, independent of $(A, B)$.*

*Consequently, any predictor $\Psi$ that depends only on the gauge-isomorphism class of radius-$r$ restricted data (i.e. $\Psi(U) = \psi([U|_Y])$ for some function $\psi$ and gauge-invariant equivalence class $[\cdot]$) is constant over $(\alpha, \beta)$ in the noiseless regime.*

*Proof.* By Lemma 12, $Y$ is contractible and does not wrap around the torus. In the noiseless construction, every face holonomy is the identity (Proposition 2), so $U^{A,B}|_Y$ is flat on $Y$. By Lemma 11, $U^{A,B}|_Y$ is gauge-equivalent to the trivial connection on $Y$, and this argument does not depend on $(A, B)$ because $Y$ contains no wrap-around.

For the second claim, if $\Psi(U) = \psi([U|_Y])$ and all $[U^{A,B}|_Y]$ coincide, then $\Psi(U^{A,B})$ is the same for all $(A, B)$, hence cannot recover $(\alpha, \beta)$. $\square$

**Connection to CTLN non-reducibility.** Theorem 6 isolates the essential difference between CTLN and local baselines: CTLN explicitly evaluates holonomy on *non-contractible* loops (global groupoid composition), which cannot be reconstructed from gauge-invariant information confined to contractible neighborhoods. In addition, CTLN uses designated face boundaries (2-cells) to form curvature features; this dependence on the cell structure is not present in methods that treat the input as only an abstract graph.

### A.7 The Categorical Nature of CTLN and Non-Reducibility to GNN

An arbitrary edge connection $U$ induces a path-holonomy functor

$$\mathcal{F}_U : \mathcal{P}(X_m^{(1)}) \to \mathbf{B}(\mathrm{SO}(3)),$$

where $\mathcal{P}(X_m^{(1)})$ is the path groupoid of the 1-skeleton and $\mathbf{B}(\mathrm{SO}(3))$ is the one-object groupoid with endomorphism group $\mathrm{SO}(3)$. Gauge transformations $g \in \mathcal{G}$ act as natural isomorphisms between these functors (Lemma 1). In the flat/noiseless regime, namely when the holonomy around every face boundary is trivial, $\mathcal{F}_U$ descends through the 2-cell relations to a functor

$$\overline{\mathcal{F}}_U : \Pi_1(X_m) \to \mathbf{B}(\mathrm{SO}(3)).$$

CTLN applies a conjugacy-invariant functional (trace/rotation angle) to a selected family of loops, so its feature map factors as

$$U \;\mapsto\; \mathcal{F}_U \;\mapsto\; \{\mathcal{F}_U(\ell)\}_{\ell \in \mathcal{L}} \;\mapsto\; \{\theta(\mathcal{F}_U(\ell))\}_{\ell \in \mathcal{L}},$$

where $\mathcal{L}$ consists of the chosen row loops, column loops, and face-boundary loops in the 1-skeleton. In the flat/noiseless regime, the row and column loops descend to representatives of the two generators of $\pi_1(T^2) \cong \mathbb{Z}^2$, while the face-boundary loops are contractible in the full cell complex $X_m$ and satisfy $H_U(\partial f) = I$. Thus the construction is naturally expressed in the language of groupoids, functors, and natural isomorphisms.

CTLN uses plaquette loops $\partial f$ to measure local curvature (Section 3.1), which is a 2-cell/cellular-sheaf type constraint: it is not a function of the 1-skeleton alone as an abstract graph unless the face structure (i.e. which 4-cycles are designated as boundaries) is provided.

We provide a structural separation result explaining why *purely local* message passing is inadequate for recovering global torus holonomies in the flat/noiseless regime.

**Theorem 7** (Global-topology obstruction for local methods on flat connections). *Fix $m$ and consider the noiseless family of flat torus connections produced by varying $(A, B)$ while keeping the vertex gauges arbitrary. Let $r < m/2$. Then the restriction of any such connection to any radius-$r$ contractible subcomplex is gauge-equivalent to the trivial connection. Consequently, any predictor that depends only on isomorphism classes of radius-$r$ neighborhoods (and is invariant under the vertex-wise gauge action) cannot distinguish different values of $(\alpha, \beta)$ in the noiseless regime.*

*Proof.* Let $U^{A,B}$ be any noiseless connection in the family and let $Y := Y_r(v)$ be any radius-$r$ induced subcomplex. Then $Y$ is contractible. Because the global connection $U^{A,B}$ is flat on all faces of $X_m$ (commutativity of $A, B$ in the generator ensures $H_{U^{A,B}}(\partial f) = I$ for all faces $f$), the restriction $U^{A,B}|_Y$ is flat on all faces of $Y$. Then there is a gauge $h \in \mathrm{SO}(3)^{V_Y}$ such that

$$(h \cdot (U^{A,B}|_Y))_e = I$$

for all edges $e$ of $Y$. Thus $U^{A,B}|_Y$ is gauge-equivalent to the trivial connection on $Y$.

For the second claim, let $\Psi$ be any predictor satisfying: (i) *$r$-locality up to isomorphism*: $\Psi(U)$ depends only on the isomorphism class of the restricted neighborhood data $U|_Y$ (for some radius-$r$ neighborhood rule), and (ii) *gauge invariance*: $\Psi(g \cdot U) = \Psi(U)$ for all vertex gauges $g$ on $X_m$. Since all restricted connections $U^{A,B}|_Y$ are gauge-equivalent to the same trivial connection on $Y$, their isomorphism classes (after quotienting by gauge) coincide. Therefore $\Psi(U^{A,B})$ is constant over all choices of $(A, B)$, hence cannot distinguish different $(\alpha, \beta)$. $\square$

Theorem 7 formalizes a key distinction: CTLN explicitly computes holonomy on *non-contractible* loops (a global, groupoid-level operation), whereas standard finite-depth message passing is inherently local and does not incorporate categorical loop composition as a primitive. This is the sense in

which CTLN is not reducible to conventional (non-categorical) CNN/GNN constructions that operate only on local neighborhoods of the 1-skeleton without explicit access to the 2-cell structure and without functorial loop evaluation.

