# OpenReview forum: "Categorical Trace Loop Networks for Gauge-Randomized Holonomy Regression"
_ICLR.cc/2026/Workshop/GRaM — ICLR 2026 Workshop GRaM Poster_

### Official Review · Reviewer_pHEM · 2026-02-08

**Rating:** 7
**Confidence:** 2

**Review:**

The authors study a learning problem with gauge-invariant data by considering learning over SO(3) for a discrete torus.
They propose Categorical Trace Loop Networks (CTLN) to learn from loop and face-based gauge invariants. Later, they also provide a theoretical justification by proving that CTLN is gauge invariant (correctness), it is stable, and it allows identifiability of torus holonomy parameters.
They conclude the paper with experiments.

pros:
- matched architecture with a motivated problem

cons:
- the paper is a bit difficult to understand even for geo-ml researchers

This is an interesting paper for gauge invariant learning, focusing on loops and faces to achieve stability and global identifiability. I suggest that the authors make their paper more accessible by adding some intuitive discussion early in the paper for an inexperienced audience.

**Pmlr Suitability:**

Yes

---

### Official Review · Reviewer_NiVV · 2026-02-17

**Rating:** 6
**Confidence:** 2

**Review:**

The paper studies a gauge-invariant prediction task, holonomy regression, where the prediction targets are group elements of $\mathrm{SO}(3)$ on a discrete torus $\mathbb Z_m \times \mathbb Z_m$. It introduces Categorical Trace Loop Networks (CTLN), which operate on gauge-invariant features to achieve gauge invariance. The core contribution is to identify those gauge-invariant features specific to this problem setup, while a simple MLP model handles the remaining regression task. CTLN achieves better performance in this task than non-gauge-invariant baselines like standard MPNN.

The paper assumes knowledge of category theory on the part of the reader, which should not be taken for granted for the audience of this workshop. The authors are encouraged to include a background/preliminary section that explains certain terminologies used in Sec 3 and 4 to make the paper more accessible.

Question:
* Is the procedure for constructing the invariant features specific to the base domain $\mathbb Z_m \times \mathbb Z_m$ considered in this paper? Can it be generalized to other domains?

**Pmlr Suitability:**

Yes

---

### Official Review · Reviewer_xswq · 2026-02-23
**Well-Motivated Gauge-Invariant Architecture**

**Rating:** 6
**Confidence:** 2

**Review:**

## Summary:
This paper studies a gauge-invariant learning problem on a discrete torus. The authors propose Categorical Trace Loop Networks (CTLN), a nwe architecture, to learn loop- and face-based gauge-invariant features derived from holonomies. Experiments show that CTLN significantly outperforms the standard message passing networks and a spectral connection-Laplacian baseline on the proposed holonomy regression task.

## Strengths:
1. This paper has a well-motivated problem setting. It clearly identifies the failure mode of standard GNNs in gauge-randomized settings.
2. The performance gap between CTLN and othr baselines is large.
3. This work builds on ideas from gauge theory, which aligns with the theme of geometry of this workshop.

## Weaknesses:
1. This paper assumes familiarity with category theory and gauge theory, which may limit accessibility for a broadeer audience.
2. The experiments focus on a single synthetic task. A discussion of broader applicability would strengthen the paper.

## Overall:
This paper is based on interesting conceptual ideas from gauge theory. While the experimental scope is limited, the contribution is appropriate for Gram.

**Pmlr Suitability:**

Yes

---

### Meta-Review · Area_Chair_H7Lo · 2026-02-25

**Decision:**

Accept

**Metareview:**

The paper presents a novel architecture for gauge-invariant data, with some theoretical justification and experiments. The topic is relevant for GRaM. The biggest issue seems to be accessibility; the paper freely uses technical terms (eg functor, holonomy) that will not be accessible to most attendees of the GRaM workshop, and the reviewers themselves all had low confidence. The reviewers are in agreement that the paper is worth accepting, but the authors should take care to ensure the camera ready and poster are accessible to a wide audience.

**Relevance To Proceedings:**

Yes — suitable for PMLR (long paper)

**Relevance To Workshop:**

Yes — suitable for GRaM

---

### Decision · Program_Chairs · 2026-03-02

Accept (Poster)